# Risk Factors of Multiple Primary Cancers Among Colorectal Cancer Survivors

**DOI:** 10.3390/cancers17132145

**Published:** 2025-06-25

**Authors:** Mulugeta Melku, Oliver G. Best, Jean M. Winter, Lauren A. Thurgood, Muktar Ahmed, Ganessan Kichenadasse, Molla M. Wassie, Erin L. Symonds

**Affiliations:** 1Flinders Health and Medical Research Institute, College of Medicine and Public Health, Flinders University, Bedford Park, SA 5042, Australia; giles.best@flinders.edu.au (O.G.B.); jean.winter@flinders.edu.au (J.M.W.); lauren.thurgood@flinders.edu.au (L.A.T.); muktar.ahmed@flinders.edu.au (M.A.); ganessan.kichenadasse@flinders.edu.au (G.K.); molla.wassie@flinders.edu.au (M.M.W.); erin.symonds@sa.gov.au (E.L.S.); 2Department of Haematology and Immunohaematology, School of Biomedical and Laboratory Sciences, College of Medicine and Health Science, University of Gondar, Gondar 196, Ethiopia; 3Medical Oncology Department, Flinders Centre for Innovation in Cancer, Flinders Medical Centre, South Adelaide Local Health Network, Bedford Park, SA 5042, Australia; 4Gastroenterology Department, Flinders Medical Centre, South Adelaide Local Health Network, Bedford Park, SA 5042, Australia

**Keywords:** colorectal cancer, multiple primary cancer, risk factors, cancer survivorship

## Abstract

Although survival rates for colorectal cancer (CRC) have improved, survivors now face a growing challenge: an elevated risk of subsequent multiple primary cancers (MPCs). This study examined factors associated with subsequent MPCs in CRC survivors. The results showed that 1 in 10 survivors—those diagnosed with CRC as their first primary cancer and who survived at least six months post-diagnosis—developed a new primary cancer. Additionally, male sex, older age, early-stage CRC at diagnosis, and loss of mismatch repair (MMR) protein expression were identified as key risk factors for MPCs. Given these findings, tailored surveillance strategies based on individual risk profiles should be implemented to enable early detection of subsequent cancers and improve long-term outcomes.

## 1. Introduction

Globally, colorectal cancer (CRC) is the third most commonly diagnosed cancer and the second cancer-related cause of death [1]. Globally, 1.93 million new cases of CRC and 0.9 million CRC deaths were reported in 2022. According to the Global Cancer Observatory reports, the number of new cases being diagnosed continues to rise, with projections of 3 million cases by 2040 and 3.6 million by 2050, which represent 55% and 85.5% increases from 2022, respectively [2,3]. Similarly, the number of new cases of CRC deaths per year is projected to increase to 1.5 million by 2040 and 1.84 million by 2050, 63.7% and 103% increases from 2022, respectively [2,4]. Despite these increases, the age-standardised rate (ASR) and mortality have declined, particularly in developed countries [5,6], likely due to the development of new treatment strategies and early detection through organised screening programs. Recent evidence shows that the ASR of CRC declined from 19.6 per 100,000 population in 2020 [7] to 18.4 per 100,000 population in 2022 [8]. Likewise, the age-standardised mortality rate (ASMR) showed a constant decline over time from 9.0 per 100, 000 population in 2020 [7] to 8.1/per 100,000 population in 2022 [8]. In addition, the global five-year survival of CRC has been shown to have improved from nearly 65% in the period of 2000–2004 to approximately 70% from the period of 2010–2014, with significant differences between developed and developing nations [9,10].

In Australia, the incidence and mortality estimates show patterns similar to the global trends. The number of reported CRC cases increased from 8911 in 1990 to 14,534 in 2020, a 63% increase from 1990 [11]. Despite the increase in the number of new cases per year, the ASR declined from 60.6 to 47.4 cases per 100,000 population between 1990 and 2020. The number of new CRC deaths also increased by 27.5% from 4283 in 1990 to 5460 in 2020, while the ASMR declined by almost half, from 29.8 deaths to 16.6 deaths per 100,000 population [12]. Moreover, the five-year survival rate improved from 60% in 1991–1995 to 71.2% for 2016–2020 [13].

With improved survivorship and decreased CRC-related deaths, there is a growing challenge of subsequent multiple primary cancers (MPCs) among individuals diagnosed with CRC [14]. According to the International Agency for Research on Cancer (IARC)/International Association of Cancer Registries (IACR), MPCs are defined as two or more cancers diagnosed in the same individual, either at the same anatomical site with different histological features or at different sites, that are not metastases, recurrences, or extensions of the index cancer [15]. Evidence has shown that CRC survivors have a significantly elevated risk of developing MPCs compared to the risk of cancer in the general population [14,16,17]. While the cancer-specific risk varied, evidence suggests that CRC survivors have a higher risk of developing MPC of the gastrointestinal tract, prostate, lung, breast, and urinary tract [16,18].

There are significant gaps that remain in understanding the effect of modifiable and non-modifiable risk factors on the development of MPC in CRC survivors. In the studies to date, male sex [18], older age at diagnosis of the index CRC [18,19], small primary tumour size [18], right-sided cancer [18], early-stage primary cancer [19], and surgical resection of the primary tumour [19] have been linked to a higher likelihood of developing subsequent malignancies. Moreover, Lee et al. (2015) reported that comorbidities such as chronic obstructive pulmonary diseases, liver cirrhosis, dyslipidaemia, and autoimmune diseases have been associated with MPCs [20]. Most of the published literature on MPCs utilise cancer registry datasets, and while these datasets are valuable for tracking long-term trends and estimating the incidence of MPCs relative to the general population, they often lack detailed information such as cancer-related variables, comorbid conditions, and treatment-related factors that may influence the occurrence of MPCs. As a result, the reported risk factors are primarily limited to socio-demographic characteristics and the site of the primary tumour. While a few studies included some clinical and treatment information, they did not include lifestyle factors, leaving evidence gaps in how modifiable and non-modifiable risk factors affect the development of MPCs. To address this, we sought to identify risk factors associated with the development of MPCs among individuals diagnosed with invasive CRC who were enrolled for a blood biomarker study at Flinders Medical Centre, South Australia.

## 2. Methods

### 2.1. Study Design, Population, Eligibility, and Outcomes

A retrospective analysis was conducted on individuals diagnosed with CRC and enrolled in a blood biomarker study at Flinders Medical Centre (Bedford Park, South Australia) who were undergoing prospective monitoring and follow-up, as previously described [21]. For the current study, adults diagnosed with all stages of invasive colorectal adenocarcinoma from January 2011 to February 2024 were included. Individuals with a history of invasive cancer prior to the CRC diagnosis, synchronous cancers (cancers diagnosed within 6 months of index CRC diagnosis), inconclusive pathology reports regarding CRC or other cancers, confirmed Lynch syndrome, and those who either died within six months of CRC diagnosis or had a follow-up period of less than six months, were excluded. Eligible study participants were followed until the date of MPC diagnosis, death, or last follow-up (30 August 2024).

In this study, according to the IARC/IACR rules, MPCs were defined as histologically distinct invasive cancers that originate either at the same site as the index CRC but with different histology or at different primary sites or tissues. These should not be extensions, recurrences, or metastases of the index CRC and must be diagnosed more than six months after the index CRC [22]. While the IARC rules for defining multiple primaries are not time-dependent, synchronous cancers (diagnosed within six months of the index CRC) were excluded to minimise detection bias, such as undiagnosed metastatic or synchronous malignancies [23]. Pathology and radiology reports were reviewed to characterise the histological and imaging features of the index CRC and subsequent cancers, to identify the type of MPC, and to rule out misclassification of CRC metastases or recurrences as MPCs.

The primary outcome of this study was to estimate the incidence of metachronous MPCs and identify the risk factors associated with their development. The secondary outcome was to assess the risk of metachronous MPCs in individuals first diagnosed with CRC and to identify the common types of MPCs that arise in this population.

### 2.2. Study Variables

Sociodemographic and behavioural characteristics: Data regarding age, sex, alcohol consumption, and smoking habits were extracted from the databases. The area-level socioeconomic status was determined using the Socio-Economic Indexes for Areas (SEIFA) data of the Australian Bureau of Statistics, which ranks areas based on the relative socioeconomic advantages and disadvantages and classifies areas into decile and percentile scales. Alcohol consumption was categorised as risky alcohol consumption if an individual reported consuming more than 10 standard drinks per week or more than four standard drinks on any single day, based on the National Health and Medical Research Council guidelines for reducing health risks from alcohol consumption [24]. Those who reported consuming 10 or fewer standard drinks per week or abstaining from alcohol were classified as “less risky or no alcohol consumption.” Regarding smoking, individuals who reported smoking at the time of enrolment in the program, as well as those who had previously smoked but had quit, were classified as current/previous smokers.

Clinical and cancer treatment variables: Height and weight at enrolment in the blood biomarker study were used to calculate the body mass index (BMI) (kg/m^2^), which was then categorised into under 25 kg/m^2^ or 25 kg/m^2^ and above. Information concerning chronic comorbidities, including diabetes mellitus, hypertension, cardiac disease, respiratory disease, and renal disease, was also collated. The index CRC tumour-related variables, such as primary site, tumour differentiation, and cancer stage based on the American Joint Committee on Cancer (AJCC 6th & 7th ed), were documented. Cancer treatment regimens, including surgical resection, chemotherapy, and radiotherapy, were compiled. Information on mismatch repair (MMR) protein expression (assessed using immunohistochemical staining) was also collected, while excluding individuals with a confirmed Lynch syndrome diagnosis and those with a family history of Lynch syndrome.

### 2.3. Statistical Analysis

The study participants were followed from the diagnosis of their index CRC until the date of MPC diagnosis, date of death, or censoring date (30 August 2024). The cumulative incidence function (CIF) was used to estimate the incidence of MPCs. The ASR was determined using a direct standardisation method against the five-year age band distribution of the South Australian population accessed from the Australian Bureau of Statistics [25], taking the average population distribution for the matched period as the standard population weight.

For comparison with the general population, the standardised incidence ratio (SIR) and 95% confidence intervals (CIs) were computed using Poisson regression by comparing the observed to the expected cases. Excess absolute risk (EAR) was calculated as the difference between observed and expected cases divided by the person-year at risk, expressed per 10,000 population [23].

To identify factors associated with metachronous MPCs, the Fine and Gray proportional sub-distribution hazard model was used, accounting for death before the development of an MPC as a competing event [19]. Model fitness was evaluated by including time-varying covariates. Both univariable and multivariable analyses were performed, with all variables with two-sided *p*-values of <0.20 in the univariable analysis included in the multivariable model. Variables with a *p*-value < 0.05 in the multivariable model were considered statistically significant. An adjusted sub-distribution hazard ratio (SHR) with a 95% confidence interval was reported as a measure of association strength. The analyses were conducted using STATA version 18 (Stata Corp LLC, College Station, TX, USA) and R statistical software version 4.4.2 (R Foundation for Statistical Computing, Vienna, Austria).

## 3. Results

### 3.1. Characteristics of the Study Participants

A total of 720 study participants, who had been diagnosed with invasive colorectal adenocarcinoma and were undergoing prospective monitoring for recurrence, were enrolled in the CRC blood biomarker study at Flinders Medical Centre. After excluding cases with prior invasive cancer (*n* = 91), cases with synchronous cancer diagnosed within 6 months of the index CRC (*n* = 29), participants with confirmed personal or family history of Lynch syndrome (*n* = 18), participants who died within 6 months of the index CRC diagnosis (*n* = 14), or participants with less than 6 months of follow-up since index CRC diagnosis (*n* = 14), 554 participants were included in the study (Figure 1).

The median age of study participants at the diagnosis of their index CRC was 65.4 years (IQR: 55.3–73.4), and 60% were male. Over half (54%) of the participants were either active smokers at the time of diagnosis or had a history of smoking, and 17% had a history of risky alcohol consumption.

The colon was the primary site of the index cancer in 65% of participants, and nearly half of the index CRC cases were diagnosed at stage I or II (48%), with the majority (83%) classified as moderately differentiated or well-differentiated tumours. Surgical resection was performed in two-thirds of the participants, and 59% received chemotherapy as part of their treatment for their index CRC. Excluding individuals with a personal or family history of Lynch syndrome (2.5% of the total population), nearly 9% of the eligible study participants had a loss of MMR protein expression, of which two-thirds had either *BRAF* mutation or *MLH1* promoter hypermethylation (Appendix A), suggesting somatic rather than germline mutation of the MMR genes. Moreover, more than half of the study participants had a history of chronic comorbid conditions, including diabetes mellitus, hypertension, and cardiac disease (Table 1).

For individuals diagnosed with an MPC, the median age at diagnosis of index CRC was 71.4 years (IQR: 66–77.5), compared to 65.5 years (IQR: 53.6–72.8) for those in the index CRC-only subset. Among the study participants with MPCs, 85% and 78% had undergone surgical resection for the index CRC and were diagnosed with their index CRC at stages I or II, respectively (Table 1).

### 3.2. Incidence, Risk, and Common Types of MPCs

Among the total study participants, 11.6% (*n* = 64) developed one or more subsequent MPCs during the 3180 person-years at risk, with a maximum of 13.6 years of follow-up. The median time to the diagnosis of a subsequent MPC after the index CRC was 5 years (IQR: 2.8–7.6 years). The most common types of MPCs were prostate cancer (21.7%), subsequent CRC (18.8%), haematological malignancies (14.5%), lung cancer (8.7%), melanoma (7.2%), and urinary tract cancers (7.2%) (Figure 2). The cumulative incidence function showed that the risk of developing MPC at 3, 5, 10, and 13.6 years of follow-up, considering death as a competing event, was 3.4% (95% CI: 2.1–5.3), 7.0% (95% CI: 4.9–9.6), 15.7% (95% CI: 12.1–19.8), and 20.2% (95% CI: 15.3–25.6), respectively. When limiting the study participants to those with a minimum of 2 or 3 years of follow-up after the index CRC, the cumulative incidence of MPC was nearly consistent with the incidence observed in participants with at least 6 months of follow-up. The cumulative incidence was 19.7% (95% CI: 14.5–25.4) for those with at least 2 years of follow-up and 19.6% (95% CI: 14.1–26.7) for those with at least 3 years of follow-up. During the follow-up time, 122 of the study participants died before developing MPCs, with a cumulative mortality of 31.8% (95% CI: 26.7–36.9%).

The crude rate of MPCs was 2026 per 100,000 population (95% CI: 2026–2588), while the ASR was 611 per 100,000 population (95% CI: 308–1365). In addition, the risk of developing MPC was significantly higher than the risk of cancer in the general population, with an SIR of 1.32 (95% CI: 1.03–1.68) and an AER of 52 cases per 10,000 population (95% 9%CI: 29–86). Similarly, the risk of developing an MPC remained significantly higher compared to the general population when limiting the study to participants with at least 2 and 3 years of follow-up after the diagnosis of index CRC, with SIRs of 1.39 (95% CI: 1.05–1.83) and 1.50 (95% CI: 1.12–2.02) for 2 and 3 years of follow-up, respectively.

### 3.3. Risk Factors Associated with MPCs

In multivariable competing risk regression, age 65 years or older at the time of index CRC diagnosis (SHR = 3.04, 95% CI: 1.57–5.95), male sex (SHR = 1.75, 95% CI: 1.01–3.019), stages I or II at the diagnosis of the index CRC (SHR = 2.20, 95% CI: 1.20–4.03), and a reduction in or loss of MMR protein expression (SHR = 2.07, 95% CI: 1.04–4.13) were significantly associated with the risk of MPC in individuals diagnosed with an index CRC (Table 2). As a sensitivity analysis, excluding participants with loss of MMR protein who did not undergo genetic testing, age, sex, and stage of CRC at diagnosis were significantly associated with MPC, while MMR protein expression was found to be marginally significant (Appendix A). The cumulative incidence plots of MPCs by sex, age, stage of CRC, and MMR protein expression are presented in Figure 3. The univariable competing risk regression results are presented in Appendix A.

## 4. Discussion

This study found that 11.5% of individuals diagnosed with invasive colorectal adenocarcinoma as their first cancer developed a subsequent metachronous MPC, resulting in an ASR of 610 cases per 100,000 population. The risk of developing MPC in this cohort was significantly higher than the expected risk in the general population, highlighting the increased cancer risk among these individuals. This study also identified factors that contribute to an elevated risk of developing subsequent metachronous MPC—specifically, male sex, older age at diagnosis of the index CRC, diagnosis with stage I or II CRC, and reduced expression of MMR proteins. These findings underscore the need for targeted surveillance and intervention strategies tailored to individual risk profiles, which would facilitate the early detection and management of subsequent cancers and ultimately improve survival outcomes.

Consistent with previous literature [16,18], the risk of MPCs is significantly elevated among CRC survivors compared to the general population. The increased risk might be attributable to several factors, including, but not limited to, overlapping mechanisms driving tumorigenesis, shared aetiology with the index cancer, genetic predisposition (other than Lynch syndrome) and long-term effects of cancer treatment [26,27]. In this study, gastrointestinal, prostate, and haematological malignancies were the most common cancers identified as MPCs among CRC survivors, which is consistent with previous literature [23]. CRC is known to share risk factors with other cancers of the gastrointestinal tract, including excessive alcohol consumption, smoking, insufficient physical activity, low-fibre diet, high cholesterol, and obesity [28]. In addition, elevated levels of inflammatory mediators can lead to chronic inflammation among CRC patients, which in turn increases cellular turnover and promotes cancer development in the gastrointestinal tract [29,30,31]. Identification of prostate cancer as a common MPC among CRC survivors is also consistent with other studies [23,32]. Both CRC and prostate cancer are age-related and common cancers, and share common risk factors, such as family history, lifestyle factors (e.g., diet and physical activity), and genetic lesions [33,34], which likely increases the risk of subsequent prostate cancer among males with an initial diagnosis of CRC. Moreover, chemotherapy and radiotherapy for the treatment of index CRC may also have long-term effects and promote tumorigenesis, particularly for cancers of the hematopoietic system [35]. Moreover, regular colonoscopy surveillance and radiological examinations for detecting recurrence and metastasis may inadvertently increase the likelihood of detecting prostate and other gastrointestinal cancers [36]. Although breast cancer is also common in the general population, it was not identified among the most common cancers arising as MPCs, primarily because most breast cancers were diagnosed either prior to or synchronously with the index CRC and were therefore excluded from the study. This is in keeping with a previous study that showed that 69% of breast cancers were diagnosed prior to CRC and that nearly 12% were diagnosed synchronously with CRC [37].

In this study, males had a two-fold higher risk of developing MPCs compared to female CRC survivors. Evidence indicates that males tend to have higher rates of lifestyle risk factors such as smoking and alcohol consumption, in addition to the effects of elevated testosterone levels [27,38,39,40]. While no statistically significant association was found between risky alcohol consumption and MPCs in this study, it is noteworthy that 73% of participants who reported risky alcohol consumption were males. Furthermore, among the participants with MPCs who reported risky alcohol consumption, the majority (86%) were also males. This suggests that risky behaviour, particularly alcohol consumption, may be more prevalent in males and may contribute to an increased risk of subsequent cancer. In addition, recent studies showed that sex disparity in cancer risks is also linked to molecular differences between males and females [41,42,43]. Yuan et al. (2016) reported that somatic mutations and copy-number alterations in genes that are frequently identified in various cancers are more prevalent in male cancer patients than in females [41]. Among these, somatic mutations of the *LKB1* gene, which encodes the major upstream kinase that activates the energy-sensing AMP-activated protein kinase pathway, and somatic copy-number alterations, such as deletion of 1p36.23 (harbouring MTOR) and 10q23.31 (harbouring PTEN), are more frequently found in male than in female cancer patients. Molecular alterations of these genes result in the loss of tumour-suppressor function, leading to uncontrolled cell growth and the development of cancer. In addition, differences in innate and adaptive immunity between males and females could also play a role in mediating the risk of cancer development [38]. The relatively balanced and controlled innate immunity, combined with a strong adaptive immune response in females, may offer some level of protection against the risk of certain cancers, particularly cancer caused by oncogenic infections [44]. This may also contribute to a better response of females to anti-cancer immunotherapy [45] and potentially reduce the risk of subsequent cancers.

In this study, older age at diagnosis of the index CRC was significantly associated with a higher risk of subsequent primary cancers, which is consistent with previous studies [32,37]. Individuals with MPCs were older than those without MPCs, with a median age of 71.4 years versus 64.5 years at diagnosis of the index CRC (*p* < 0.001). Similarly, Halamkova et al. (2021) reported that individuals with second primary cancers (SPCs) were older than those without SPCs and found that older age increased the odds of developing SPCs [37]. Several factors may contribute to the higher risk of subsequent cancers in older individuals. First, aging of the immune system, or immunosenescence, makes it less efficient at detecting and eliminating abnormal or pre-malignant cells, leading to an accumulation of somatic mutations and, ultimately, genetic alterations that drive tumorigenesis [46]. Secondly, continued exposure to cancer-causing environmental and lifestyle-related risk factors, such as smoking, alcohol, unhealthy diet, and occupational hazards, which may have contributed to the development of the index CRC, might continue after diagnosis and lead to an increased risk of additional cancers [47,48]. Thirdly, with increased survival rates, older individuals may now live long enough to develop a second primary cancer, and fourthly, certain treatments for CRC, such as radiation therapy or chemotherapy, may increase the risk of secondary cancers, particularly among long-term survivors and older individuals, as aging reduces the ability to respond to and repair cancer-causing, treatment-induced cellular damage. Although evidence suggests that radiotherapy exposure has a longer latency period for cancer risk than many chemotherapeutic regimens [49], the risk of subsequent cancers may be accelerated among older individuals.

In this study, we observed a significant association between early-stage index CRC at diagnosis and development of MPCs. Patients with early-stage CRC generally have better prognoses and longer survival times, during which they may develop an MPC. In contrast, individuals diagnosed with advanced-stage CRC tend to have poorer prognoses and shorter survival durations, reducing the likelihood of developing MPCs. This finding is consistent with Yang et al. (2017), who reported that patients with early-stage CRC had a greater likelihood of developing a subsequent CRC [50]. These results highlight the importance of long-term surveillance in survivors of early-stage CRC to facilitate early detection and intervention for MPCs. In addition, individuals diagnosed with early-stage CRC generally have a low risk of recurrence [51], and they often undergo less intensive follow-up and surveillance, including screening for subsequent MPCs. While reduced monitoring is appropriate given the favourable prognosis of the index cancer, it may lead to delayed detection of MPCs, increasing the likelihood of their diagnosis at a more advanced stage [52].

In this study, the reduction in or loss of MMR protein expression was also associated with an increased risk of MPCs. Following exclusion of individuals with a confirmed personal or family history of Lynch syndrome, those with reduced or lost expression of one or more MMR proteins had a two-fold higher risk of developing subsequent MPCs compared to those with normal expression levels. Evidence suggests that up to 15% of CRC cases exhibit microsatellite instability (MSI), the majority of which are sporadic and mostly resulting from MMR protein deficiency, primarily caused by epigenetic silencing of the *MLH1* gene via promoter hypermethylation or somatic mutations in MMR genes [53]. MMR proteins, including MLH1, MSH2, MSH6, and PMS2, play a crucial role in maintaining genomic stability by correcting DNA replication errors, such as base mismatches and insertion–deletion loops [54]. MSI-CRC tumours frequently harbour mutations in tumour suppressor genes such as *KRAS* and pro-oncogenic genes, including *BRAF*; the V600E mutation in *BRAF* in particular is characteristic of sporadic MSI-CRC [55]. MMR protein deficiency promotes mutagenesis in tissues beyond the primary tumour site, potentially increasing the risk of additional malignancies in the bowel and other gastrointestinal organs [56]. Interestingly, among the participants in the current study, who experienced MPCs and had reduced or absent MMR protein expression, 44% developed metachronous CRC. Furthermore, a recent study demonstrated that the mutational landscapes, in terms of mutation signatures, genome-wide mutation densities, and microsatellite instability profiles, of hereditary and sporadic MSI-CRCs are highly similar. However, cases of sporadic MSI-CRC have significantly higher expression of immune-related genes [57], suggesting that patients with sporadic MSI-CRC who exhibit a loss of MMR proteins may be at an increased risk of developing subsequent malignancies. In general, the loss of MMR proteins in sporadic CRC leads to defective DNA repair, resulting in increased genetic instability and a high mutation rate, known as a mutator phenotype [58]. This elevated mutational burden generates a substantial neoantigen load, which may trigger an immune response but can eventually promote immune evasion [59]. Together, these factors contribute to a higher risk of developing multiple primary cancers in the colorectum and other organs.

### Limitations and Strengths of the Study

The limited availability of data related to factors such as diet, physical activity, and occupational and environmental exposure, which are known to have a significant impact on cancer development, was a limitation in this study. In addition, as the number of study participants who developed MPC were relatively small, this limited the ability to perform more detailed stratified analyses based on specific characteristics of the participants, including histological subtype, as well as treatment-related factors such as radiation dose, type of surgery, and type of systemic therapy received. Despite these limitations, the study was able to explore factors associated with the risk of MPC development, including sociodemographic, clinical, and tumour-related variables. The other strength of the study is the use of statistical methods that account for competing events, as a substantial portion of study participants may not survive their index CRC or succumb to other diseases before developing MPCs. Therefore, we employed the CIF and competing risk regression analysis, with death considered as a competing event. This approach could provide more accurate estimates of MPC occurrence over time and risk factors associated with MPCs.

## 5. Conclusions

Individuals diagnosed with CRCs are at a higher risk of developing subsequent cancers than the general population. Older age, male sex, early-stage tumour diagnosis, and tumours with a reduction in or loss of expression of the MMR proteins are factors that are associated with an increased risk of MPCs. These findings underscore the importance of implementing targeted surveillance and intervention strategies tailored to an individual’s specific risk profile. Employing strategies that identify high-risk groups may help in the early detection and management of CRC patients who are the most vulnerable to the development of MPCs, ultimately improving outcomes and reducing the burden of these cancers. Moreover, prospective studies with larger sample sizes, incorporating lifestyle, environmental, clinical, and other relevant factors, are essential for comprehensively identifying the sociodemographic, socioeconomic, behavioural, clinical, and cancer-related factors associated with MPCs among CRC survivors.

## Figures and Tables

**Figure 1 cancers-17-02145-f001:**
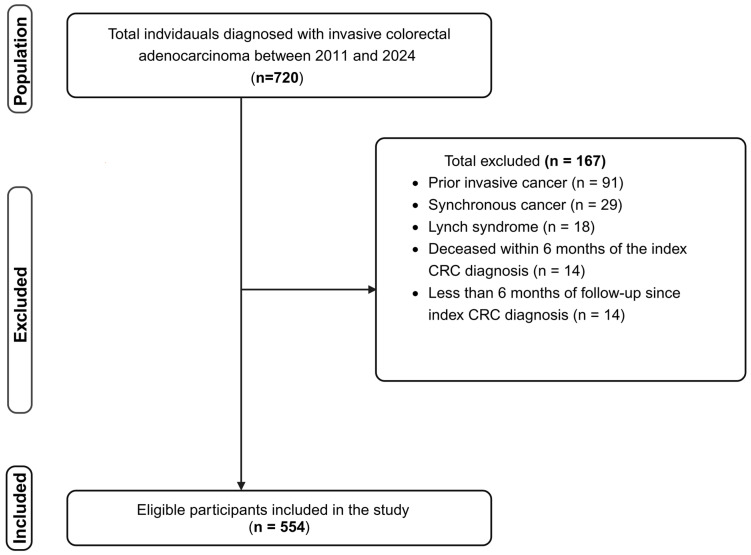
Flowchart of study participant selection among individuals diagnosed with CRC at Flinders Medical Centre. CRC: colorectal cancer.

**Figure 2 cancers-17-02145-f002:**
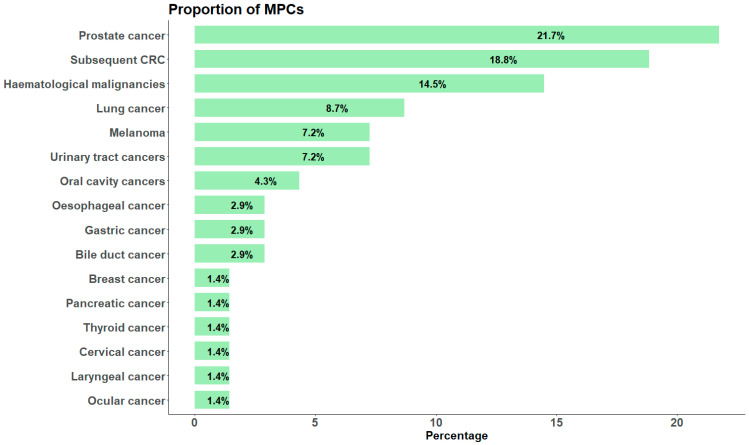
The different types and proportions of the cancers that arose as MPCs among individuals diagnosed with an index CRC. CRC: colorectal cancer; MPCs: multiple primary cancers.

**Figure 3 cancers-17-02145-f003:**
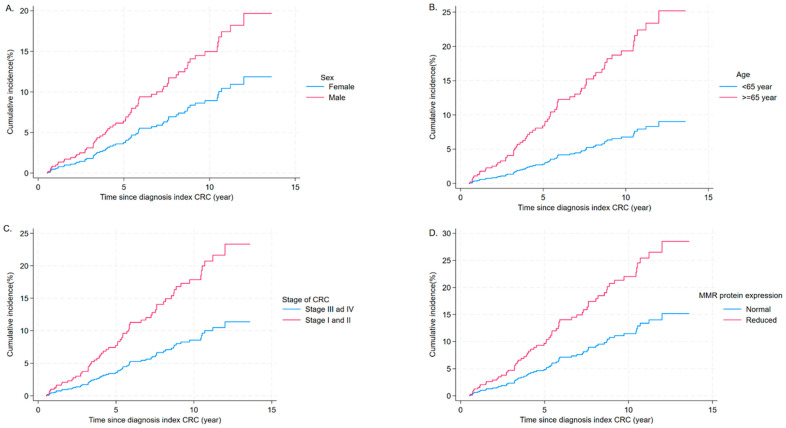
The cumulative incidence of MPC among CRC survivors by study participants’ characteristics, estimated using competing risk regression. (**A**) By sex, (**B**) by age at the diagnosis of index CRC, (**C**) by stage of index CRC at diagnosis, and (**D**) by the expression of mismatch repair protein, as detected by immunohistochemical staining. CRC: colorectal cancer; MMR: mismatch repair.

**Table 1 cancers-17-02145-t001:** Sociodemographic, behavioural, and clinicopathological characteristics of individuals diagnosed with the index CRC alone or index CRC and a subsequent MPC.

Characteristics		Total (*n* = 554)	Index CRC Only (*n* = 490)	MPC (*n* = 64)	*p*-Value
Age at diagnosis of the index CRC (years)	Median (IQR)	65.4 (55.3–73.4)	64.5 (53.6–72.8)	71.4 (66.1–77.5)	**<0.001** *
Sex	Male	332 (59.9%)	287 (58.5%)	45 (70.3%)	0.070
	Female	222 (40.1%)	203 (41.4%)	19 (29.7%)	
Socioeconomic status (quintile)	Lowest to low	286 (51.6%)	149 (50.8%)	37 (57.8%)	0.292
	Middle to highest	268(48.4%)	241(49.2%)	27 (42.2%)	
Smoking habit	Current/previous smoker	299(54.0%)	264 (53.9%)	35 (54.7%)	0.903
	No smoking	255 (46.0%)	226 (46.1%)	29 (45.3%)	
Alcohol consumption	Risky alcohol consumption	93 (16.8%)	78 (15.9%)	15 (23.4%)	0.130
	Less risky or no alcohol consumption	461 (83.2%)	412 (84.1%)	49 (76.6%)	
Site of index CRC	Colon	360 (65.0%)	310 (63.3%)	50 (78.1%)	**0.019**
	Rectum	194 (35.0%)	180 (36.7%)	14 (21.9%	
Tumour differentiation	Moderately to well differentiated	459 (82.9%)	407 (83.1%)	52 (81.2%)	0.718
	Poorly differentiated	95 (17.1%)	83 (16.9%)	12 (18.8%)	
Stage of CRC at diagnosis	Stage I and II	264 (47.7%)	219 (44.7%)	45 (70.3%)	**<0.001**
	Stage III and IV	290 (52.3%)	271 (55.3%)	19 (29.7%)	
MMR protein expression by IHC	Loss	49 (8.8%)	40 (8.2%)	9 (14.1%)	0.118
	Normal	505 (91.2%)	450 (91.8%)	55 (85.9%)	
Surgery for index CRC	Yes	381 (68.6%)	327 (66.7%)	54 (84.5%)	**0.004**
	No	173 (31.4%)	163 (33.3%)	10 (15.6%)	
Chemotherapy for index CRC	Yes	324 (58.5%)	301(61.4%)	23 (35.9%)	**<0.001**
	No	230 (41.5%)	189 (38.6%)	41 (64.1%)	
Radiotherapy for index CRC	Yes	143 (25.8%)	135 (27.6%)	8 (12.5%)	**0.01**
	No	411 (74.2%)	355 (72.4%)	56 (87.5%)	
BMI	<25 kg/m^2^	152 (27.4%)	135 (27.6%)	17 (26.6%)	0.868
	≥25 kg/m^2^	402 (72.6%)	355 (72.4%)	47 (73.4%)	
Hypertension	Yes	259 (46.8%)	220 (44.9%)	39 (60.4%)	**0.016**
	No	295 (53.2%)	270 (55.1%)	25 (39.1%)	
Cardiac diseases	Yes	136 (24.5%)	113 (23.0%)	23 (35.9%)	**0.024**
	No	418 (75.5%)	377 (76.9%)	41 (64.1%)	
Chronic renal disease	Yes	29 (5.2%)	27 (5.5%)	2 (3.1%)	0.561 ^#^
	No	525 (94.8%)	463 (94.5%)	62 (96.9%)	
Diabetes mellitus	Yes	102 (18.4%)	87 (17.8%)	15 (23.4%)	0.270
	No	452 (81.6%)	403 (82.2%)	49 (76.6%)	
Chronic respiratory disease	Yes	127 (22.9%)	109 (22.2%)	18 (28.1%)	0.293
	No	427 (77.1%)	381 (77.8%)	46 (71.9%)	

BMI: body mass index; CRC: colorectal cancer; IHC: immunohistochemistry; IQR: interquartile range; Kg: kilogram; MMR: mismatch repair; m^2^: metre square; MPC: multiple primary cancer. * An independent t-test was used to compare the mean value between groups for continuous variables. Pearson’s chi-square and ^#^ Fisher’s exact test were employed to analyse the association between categorical variables. Bold *p*-values indicate significance at *p* < 0.05.

**Table 2 cancers-17-02145-t002:** Univariable and multivariable competing risk regression of MPCs in individuals diagnosed with index colorectal cancer.

Variables	Categories	Univariable Analysis		Multivariable Analysis	
		SHR (95% CI)	*p*-Value	SHR (95% CI)	*p*-Value
Age at index CRC diagnosis, years	<65	1.0		1.0	
	≥65	3.61 (2.00, 6.49)	<0.001	**3.04 (1.56, 5.95)**	0.001
Sex	Female	1.0		1.0	
	Male	1.44 (0.85, 2.45)	0.181	**1.75 (1.01, 3.01)**	0.045
Site of index CRC	Rectum	1.0		1.0	
	Colon	1.82 (1.01, 3.)	0.047	0.90 (0.35, 2.37)	0.837
Stage of index CRC at diagnosis	Stage III and IV	1.0		1.0	
	Stage I and II	2.34 (1.37, 3.99)	0.002	**2.20 (1.20, 4.03)**	0.011
Expression of MMR protein(s)	Normal	1.0		1.0	
	Loss	2.47 (1.23, 4.96)	0.011	**2.08 (1.04, 4.13)**	0.037
Surgery for index CRC	No	1.0		1.0	
	Yes	2.03 (1.05, 3.93)	0.036	1.01 (0.33, 3.10)	0.989
Chemotherapy for index CRC	Yes	1.0		1.0	
	No	2.07 (1.24, 3.45)	0.005	0.83 (0.44, 1.56)	0.559
Radiotherapy for index CRC	Yes	1.0		1.0	
	No	2.52 (1.20, 5.30)	0.014	1.98 (0.49, 8.06)	0.340
Hypertension	Yes	1.0		1.0	
	No	1.75 (1.06, 2.89)	0.028	1.08 (0.61, 1.91)	0.797

CRC: colorectal cancer; CI: confidence interval; MMR: mismatch repair; SHR: sub-distribution hazard ratio. Bold values indicate statistical significance in multivariable analysis at *p*-value <0.05.

## Data Availability

The data used to prepare this manuscript are available within the manuscript and its Appendix A. Additional information can be provided upon reasonable request to the corresponding author.

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
