# Peer review of "Risk Factors of Multiple Primary Cancers Among Colorectal Cancer Survivors"

_cancers, 2025, doi:10.3390/cancers17132145_

Round 1

Reviewer 1 Report

Comments and Suggestions for Authors

Dear Authors,

Retrospective analysis describes well that – adults diagnosed with Invasive colorectal adenocarcinoma at Flinders Medical Centre since 2011, who had at least 6 months of post-CRC follow-up. Sociodemographic factors, clinical information, tumour characteristics, and treatment types were collected.

Individuals diagnosed with CRCs are at a higher risk of developing subsequent cancers than the general population. Older age, male sex, early-stage tumour diagnosis, and tumours with reduced or loss of expression of the MMR proteins are factors that are associated with an increased risk of MPCs.

The following steps should provide more clear information for readers to enjoy it

Minor revisions are required

Abstract section

1) Mention number of CRC patients’ are participated and data collection years. 

Results section

1) Table 1. Sociodemographic, behavioural, and clinicopathological characteristics of individuals diagnosed with the index CRC alone or index CRC and a subsequent MPC – Stage of CRC at diagnosis – mention Stage I, II, III and IV analysis individually, not combined.

2) Interesting results in Figure 2. The different types and proportions of the cancers that arose as MPCs among individuals diagnosed with an index CRC – Please keep a subtitles – mention more information like prostate, blood and lung cancers.

Author Response

  1. Point-by-point response for Reviewer #1 comments

Reviewer #1

Comments and Suggestions for Authors

Dear Authors,

Retrospective analysis describes well that – adults diagnosed with Invasive colorectal adenocarcinoma at Flinders Medical Centre since 2011, who had at least 6 months of post-CRC follow-up. Sociodemographic factors, clinical information, tumour characteristics, and treatment types were collected.

Individuals diagnosed with CRCs are at a higher risk of developing subsequent cancers than the general population. Older age, male sex, early-stage tumour diagnosis, and tumours with reduced or loss of expression of the MMR proteins are factors that are associated with an increased risk of MPCs.

The following steps should provide more clear information for readers to enjoy it

Minor revisions are required

Authors’ response:

Thank you for the critical review and the insightful comments and suggestions. We have revised the manuscript accordingly and provided a point-by-point response below.

Abstract section

  • Mention number of CRC patients’ are participated and data collection years. 

Authors’ response:

The total study consisted of 554 individuals diagnosed with invasive adenocarcinoma between 2011 and 2024, and this information has now been added to the abstract (Page 1, lines 33-37) of the revised manuscript. The additional text reads:

“Of the total 554 eligible study participants, 12% developed MPC, with a median follow-up time of 5 years (interquartile range: 2.8 - 7.6 years) until the diagnosis of MPC.”

We also revised the Methods section on page 3, line 113, to detail the study period. The new text reads:

 “……. adults diagnosed with all stages of invasive colorectal adenocarcinoma from January 2011 to February 2024 were included.” 

Results section

  • Table 1. Sociodemographic, behavioural, and clinicopathological characteristics of individuals diagnosed with the index CRC alone or index CRC and a subsequent MPC – Stage of CRC at diagnosis – mention Stage I, II, III and IV analysis individually, not combined.

Authors’ response:

We presented the data and the subsequent statistical analysis by categorising stages I and II as one group and stages III and IV as another group for two reasons: 1) Stages I and II typically represent localised disease (early stage) with similar management strategies (primarily surgical) and better survival outcomes compared to Stages III and IV (late stage), which often involve lymph node or distant metastases, requiring more aggressive multimodal treatments; 2) The number of multiple primary cancers (MPCs) in our study was relatively small (n=64), and further stratification into four individual cancer stages would reduce the statistical power, yielding more unreliable estimates of risk factors. Additionally, in the original submission version, we acknowledged that this is a limitation of the study and have recommended that future research consider incorporating more behavioural, clinical, treatment, and cancer-related factors using larger-sized, prospective study designs (refer to page 12, lines 403 -429).

  • Interesting results in Figure 2. The different types and proportions of the cancers that arose as MPCs among individuals diagnosed with an index CRC – Please keep a subtitles – mention more information like prostate, blood and lung cancers.

Authors’ response:

In the original submission version, we have detailed the types and proportions of the different MPCs that arose among individuals diagnosed with an index CRC, as outlined in the section titled “Incidence, Risk, and Common Types of MPCs,” as shown in Figure 2. In addition to prostate cancer, lung cancer, and haematological malignancies (e.g., leukemias and lymphomas), we have the statement in the Results section to include melanoma and urinary tract cancers in the revised manuscript. Please refer to page 7, lines 227–229, for further details. The new text now reads:

The most common types of MPCs were prostate cancer (21.7%), subsequent CRC (18.8%), haematological malignancies (14.5%), lung cancer (8.7%), melanoma (7.2%) and urinary tract cancers (7.2%) (Figure 2)”

Kindly

Mulugeta Melku

Reviewer 2 Report

Comments and Suggestions for Authors

Although the survival rate of colorectal cancer patients has improved in recent years, patients with a history of colorectal cancer may develop several cancers in their lives. A better understanding of MPCs' risk factors may be beneficial for cancer management. However, the aetiology behind multiple primary cancers remains unclear. In this context, this manuscript aims to determine the incidence and identify risk factors associated with metachronous multiple primary cancers among colorectal cancer survivors.

It is a well-conducted study associated with robust statistical analysis. Thus, I consider that the paper would be well-received by peers in the field. As a consequence, I recommend this manuscript as a suitable manuscript for publication in the Cancers journal, with a few minor suggestions:

-             Kindly indicate in Table 1 the p-values obtained with Fisher’s exact test.

-             An additional figure to illustrate the mechanism of MMR protein expression loss and the higher risk of multiple primary cancer development may highlight this part of the Discussion section.

-             The authors should add supplementary references in lines 54 and 320 (’’ Recent studies showed that ...’’ - it is more than one study)

-  Please revise the reference list according to the ''Cancers'' journal recommendations.

Author Response

Point-by-point response for Reviewer #2 comments

Reviewer #2

Comments and Suggestions for Authors

Although the survival rate of colorectal cancer patients has improved in recent years, patients with a history of colorectal cancer may develop several cancers in their lives. A better understanding of MPCs' risk factors may be beneficial for cancer management. However, the aetiology behind multiple primary cancers remains unclear. In this context, this manuscript aims to determine the incidence and identify risk factors associated with metachronous multiple primary cancers among colorectal cancer survivors.

It is a well-conducted study associated with robust statistical analysis. Thus, I consider that the paper would be well-received by peers in the field. As a consequence, I recommend this manuscript as a suitable manuscript for publication in the Cancers journal, with a few minor suggestions:

Authors’ response:

Thank you for the critical review and the insightful comments and suggestions. We have revised the manuscript accordingly and provided point-by-point responses below.

  1. Kindly indicate in Table 1 the p-values obtained with Fisher’s exact test

Authors’ response: Thank you for the comment. We have now indicated the p-value with a symbol in Table 1 of the revised manuscript (Results section, Page 6, line 220-222). 

  1. An additional figure to illustrate the mechanism of MMR protein expression loss and the higher risk of multiple primary cancer development may highlight this part of the Discussion section.

Authors’ response: Our study findings show there is an association with MMR loss and a higher risk of subsequent primary cancer, but we recognise that this association may not be causative, and we are cautious not to overstate this association by hypothesising concerning the mechanisms of action. However, we also recognise that further discussion of this potential association is warranted.  Therefore, we have now expanded on how MMR protein loss may be linked to an increased risk of subsequent cancer in cases of sporadic CRC (Discussion, pages 11-12, lines 395-401). The new text reads:

In general, the loss of MMR proteins in sporadic CRC leads to defective DNA repair, resulting in increased genetic instability and a high mutation rate, known as a mutator phenotype. This elevated mutational burden generates a substantial neoantigen load, which may trigger an immune response but can eventually promote immune evasion. Together, these factors contribute to a higher risk of developing multiple primary cancers in the colorectum and other organs.”

  1. The authors should add supplementary references in lines 54 and 320 (’’ Recent studies showed that ...’’ - it is more than one study)

Authors’ response: Additional relevant references have now been cited to support this sentence. Please refer to page 2, lines 54-57 of the Introduction and page 10, lines 320-321 of the Discussion in the revised manuscript.   

  1. Please revise the reference list according to the ''Cancers'' journal recommendations.

Authors’ response: We have updated the reference list using the MDPI journal EndNote style. 

Sincerely,

Mulugeta Melku